# Perioperative Pain Management for Mastectomy in Dogs: A Narrative Review

**DOI:** 10.3390/ani15091214

**Published:** 2025-04-25

**Authors:** Giada Giambrone, Giuseppe Catone, Gabriele Marino, Alessandra Sfacteria, Renato Miloro, Cecilia Vullo

**Affiliations:** 1Department of Veterinary Sciences, University of Messina, Via G. Palatucci, 98168 Messina, Italy; giuseppe.catone@unime.it (G.C.); gabriele.marino@unime.it (G.M.); alessandra.sfacteria@unime.it (A.S.); renato.miloro@gmail.com (R.M.); 2Department of Chemical, Biological, Pharmaceutical and Environmental Sciences, University of Messina, Viale Ferdinando Stagno d’Alcontres 31, 98166 Messina, Italy; cecilia.vullo@unime.it

**Keywords:** mastectomy, pain management, dog

## Abstract

Pain management is crucial in veterinary oncology. Mastectomy is the treatment of choice for canine mammary tumours and is related to pain from moderate to severe. This narrative review aims to resume and analyse analgesic techniques and drugs reported in the literature for mastectomy in dogs. Systemic analgesics include non-steroidal anti-inflammatory drugs (such as meloxicam and robenacoxib), opioids (tramadol, methadone, fentanyl) and local anaesthetics (lidocaine, bupivacaine). Techniques such as epidural blockade, local infiltration and tumescent infiltration anaesthesia improve pain management and reduce the need for opioid drugs. Alternative methods such as electroacupuncture may reduce the need for analgesics. In conclusion, although a multimodal approach, combining different analgesic strategies, may minimise side effects and improve postoperative recovery, the choice of a single drug bolus followed by continuous infusion may be an effective strategy in order to manage intra- and postoperative pain.

## 1. Introduction

Mammary gland tumours are the most common hormone-dependent neoplasia in female dogs [1]. Canine mammary tumour development risk increases with age [2,3,4,5]. Mastectomy remains the treatment of choice for many mammary tumours [6] and different surgical procedures may be performed [7]. Although mastectomy in dogs is associated with a relatively low morbidity, it is considered an invasive surgery that may cause moderate to severe pain [8]. Aggressive unilateral and radical mastectomy is associated with longer surgical times, higher nociception, and postoperative stress, compared with regional mastectomy [9]. Even if surgical excision in veterinary oncology is considered the gold standard for many tumour types, there is a residual risk of intra-surgical dissemination of tumour cells. The main risk of tumour dissemination occurs in the perioperative period and is related to cellular and immune changes influenced by stress and pain [10]. In veterinary medicine, pain recognition and management are challenges since pain is a complex multidimensional phenomenon to which the animals respond with physiological and behavioural reactions [11,12]. Pain management in the perioperative period may also influence protumoral and anti-immune mechanisms [10]. Various modalities and drugs are used to manage pain in patients. Recently published pain management guidelines emphasise that adequate pain control often requires a multimodal approach [13,14].

The perioperative use of opioids is a traditional practice in oncology [14]. However, opioid agonists are related to many different side effects in cancer patients, such as immunosuppression, stimulation of neoplastic cell growth, and increased risk of tumour metastasis [15,16,17,18]. The multimodal analgesic plans incorporate drugs or therapies that act on different areas of the pain pathway and work synergistically using lower doses of each analgesic drug, thus limiting adverse effects [19,20].

Considering the importance of effective analgesia for pain management during mastectomy, this review aims to evaluate studies published between 2001 and 2024 regarding the different drugs applied and techniques performed in bitches undergoing mastectomy to improve perioperative analgesia.

## 2. Materials and Methods

A literature search was conducted for manuscripts relating to the perioperative pain treatment in mastectomy in dogs. A comprehensive literature search was performed for manuscripts relating to dog mastectomy from 1 January 2001 to 20 July 2024. NCBI-PubMed, Web of Science, and SciVerse Scopus were used exhaustively as databases. For all three databases, the search strings were meticulously crafted and researched using the following terms: [(dog OR dogs) AND (mastectomy OR regional mastectomy OR unilateral mastectomy OR bilateral mastectomy) AND (analgesia OR antinociception OR pain)]. Excluded were single case reports, review articles, simulations or cadaveric studies, editorials, proceedings, meeting abstracts, articles not written in English. Papers focused on surgical approaches and techniques with marginal information about analgesia and pain management were also excluded. The results are organised into a narrative review with different sections depending on the analgesic administration techniques (Figure 1).

## 3. Systemic Analgesia

Papers included in this review evaluating system analgesic protocols in dogs undergoing mastectomy are summarised in Table 1.

### 3.1. Systemic Administration

Meloxicam is an injectable cyclooxygenase-2 (COX-2) selective nonsteroidal anti-inflammatory drug (NSAID) extensively used in small animal medicine. Nakagawa et al. [21] investigated the pre-emptive treatment with meloxicam (0.2 mg/kg SC) and butorphanol (0.1 mg/kg IV) administered 2 h before mastectomy compared with a control group that received butorphanol only just before surgery. Cardiovascular parameters were monitored for 14 days, and renal parameters and serum cortisol levels were monitored for 24 h after surgery. The results showed that perioperative administration of meloxicam reduced unfavourable postoperative changes in the cardiovascular system, such as hypertension, without influencing renal function. Although the two groups showed no significant differences in serum cortisol, the cortisol level in the group treated with meloxicam was lower from 4 to 24 h after mastectomy [21]. Meloxicam was also associated with gabapentin, a non-opioid drug with an anticonvulsant and antinociception role, in dogs undergoing mastectomy [22]. Indeed, this association has shown a significant opioid-sparing efficacy considering the role of inhibiting peripheral and central nervous system prostaglandin synthesis and reducing the hyperexcitability of dorsal horn neurons following tissue damage, respectively, for NSAIDs and gabapentin [23,24]. In dogs undergoing mastectomy, a group was treated with gabapentin (10 mg/kg PO), administered 120 min before surgery and continued for three days, and meloxicam (0.2 mg/kg IV) before surgery and daily doses for up to 10 days, while in another group, gabapentin was substituted by a placebo solution. Although the pain scores with the Glasgow Composite Measure Pain Scale (GCMPS) and Visual Analogue Scale (VAS) showed no significant differences, dogs in the gabapentin group had a significantly lower requirement of rescue doses of morphine by 44% [22]. Similarly, robenacoxib, another COX-2 selective NSAID, is effective in the control of postoperative pain. In a study, robenacoxib (0.2 mg/kg SC) administered 30 min before surgery and for two additional days showed a significant decrease in the rescue analgesia requirement with a considerable reduction in postoperative pain in different soft tissue surgeries in dogs, including mastectomies [25].

Texeira et al. [26] investigated the effects of tramadol alone or in combination with meloxicam or dipyrone on postoperative pain in twenty-seven bitches undergoing unilateral mastectomy. Tramadol is an opioid that leads to fewer side effects and has longer-lasting effects than morphine [27,28,29]. Dipyrone is an NSAID with weaker COX inhibitor activity compared to meloxicam [30]. Dogs undergoing mastectomy were assigned to receive tramadol (3 mg/kg IV) alone or combined with dipyrone (30 mg/kg IV) or meloxicam (0.2 mg/kg IV), and at 8 and 16 h after extubation received additional doses of tramadol (for groups of tramadol alone or tramadol–meloxicam) or tramadol with dipyrone (for the group of tramadol–dipyrone). Rescue analgesia was administered in six dogs that received tramadol only, four dogs that received tramadol plus dipyrone, and two dogs that received tramadol plus meloxicam. There were no significant differences between groups in postoperative pain scores, assessed with VAS and GCMPS, with little benefit using the association of tramadol with meloxicam or dipyrone. Indeed, in dogs that received treatment with one of the two NSAIDs, the vocalism frequency upon waking and the time taken to reposition in sternal recumbency decreased. The study concluded that tramadol alone or in combination with NSAIDs provided adequate analgesia for 24 h [26]. However, tramadol alone was inadequate for pain control in a study by Uscategui et al. [31]. Dogs treated with methadone (0.5 mg/kg IM) 10 min before anaesthesia induction showed significantly lower rescue analgesia requirements and pain scores, assessed with the Melbourne scale [32], compared to dogs that received tramadol (5 mg/kg IM) [31]. Even if pre-emptive administration of methadone provided better pain management after surgery, it was not completely adequate in controlling pain in dogs undergoing ovariohysterectomy and unilateral mastectomy [31]. Additionally, methadone (0.5 mg/kg IM) administration twenty minutes before or during surgery showed a better analgesic effect than postoperative administration with a lower requirement of rescue analgesia [33]. Methadone decreased the dose of propofol required for induction and reduced the need for analgesics while keeping blood pressure stable. This highlights methadone’s potential in multimodal pain management, considering the risk of clinical hypercapnia as a side effect requiring special ventilation care [31,33]. However, regarding the use of tramadol as an analgesic, a systematic review by Donati and coworkers on the efficacy of tramadol for postoperative pain management in dogs concluded that tramadol administration is likely to result in a reduced need for rescue analgesia versus no treatment or placebo or compared with buprenorphine, codeine, and nalbuphine. Tramadol may also increase the need for rescue analgesia compared with methadone and COX inhibitors [34].

Ketamine was evaluated to improve postoperative opioid analgesia and feeding behaviour. In a study by Sarrau et al. [35], twenty-seven bitches were divided into three groups after surgery was performed. The placebo group received a bolus of morphine chlorhydrate (0.1 mg/kg IV) and a bolus of isotonic saline (0.09 mg/kg IV) followed by a six-hour constant rate infusion (CRI) of isotonic saline (0.5 mL/kg/h). The low-dose ketamine group received a bolus of morphine, a bolus of ketamine (150 µg/kg) diluted in isotonic saline followed by a six-hour CRI of ketamine at a dose of 2 µg/kg/min diluted in isotonic saline to permit a rate of 0.5 mL/kg/h. The high-dose ketamine group received an analogue protocol, a dose of ketamine in a bolus of 700 µg/kg and a six-hour CRI of 10 µg/kg/min. Ketamine administration irrespective of dosage showed no better results than the placebo group on postoperative pain, assessed throughout VAS and the French Veterinary Association for Anaesthesiology and Analgesia pain scoring scale (4A VET pain scale) [36] and on sedation scores assessed with the Young et al. scale [35,37]. However, the high-dose ketamine group enhanced food intake, improving patient feeding behaviour [35]. In another study, the association of ketamine, lidocaine and maropitant led to an improvement in pain management [38]. Specifically, a group of dogs received a ketamine bolus (1 mg/kg), lidocaine bolus (1.5 mg/kg), maropitant bolus (1.5 mg/kg IV) and continuous infusion of ketamine, lidocaine and maropitant (10 µg/kg/min, 50 µg/kg/min and 100 µg/kg/h, respectively) initiated at the start of surgery and maintained until one hour postoperatively and another group received the same protocol without maropitant. The levels of analgesia were evaluated at specific time points using a simple descriptive scale, visual interactive and dynamic analogue scale (DIVAS), numerical classification scale (NCS), and short-form Glasgow composite pain scale (GCMPS-SF). The dog group receiving maropitant required fewer rescues with a significant reduction in postoperative pain with minimal cardiorespiratory effects [38]. This is related to the role of maropitant as an adjuvant agent that inhibits substance P, a neuropeptide released after tissue injuries, thereby reducing inflammation and alleviating hyperalgesia [39,40], although in human medicine, neurokinin-1 receptor antagonists, such as maropitant, have failed to provide clinical analgesia, and there is very little evidence-based information on the putative clinical use of maropitant for pain and inflammation [41,42]. The association of ketamine (bolus 0.5 mg/kg and intra- and postoperative CRI 20 μg/kg/min) and fentanyl (bolus 5 μg/kg and intraoperative CRI 5 μg/kg/h and postoperative CRI 2 μg/kg/h) showed no significant differences when compared to groups treated with one of the two drugs. Specifically, ketamine, fentanyl or their association did not cause cardiovascular changes, with satisfactory intraoperative analgesia and no differences in postoperative pain, assessed throughout GCMPS-SF and the mechanical nociceptive threshold (MNT). However, the addition of fentanyl to ketamine treatment increased plasma ketamine concentration with potential for analgesia during the 8 h of administration [43]. Another study confirmed that fentanyl (loading dose of 5 µg/kg followed by a CRI of 9 µg/kg/h), alone or combined with lidocaine (loading dose of 2 mg/kg followed by 3 mg/kg/h CRI) and ketamine (loading dose 1 mg/kg followed by 0.6 mg/kg/h CRI), was effective in the intraoperative pain management. However, dogs treated with fentanyl associated with ketamine and lidocaine did not require any postoperative rescue analgesia with lower postoperative pain, assessed throughout GCMPS-SF [44]. The decreased postoperative hyperalgesia is probably related to the action at different nociceptive pathways of ketamine, lidocaine and fentanyl [45,46]. All dogs included in the study were premedicated with acepromazine (0.02 mg/kg IV) and morphine (0.5 mg/kg IV) and received meloxicam (0.1 mg/kg) at the closure of the surgical wound. In both study groups, transient arterial hypotension after loading dose was the most concerning adverse effect, most notably in the group that received the association of the three drugs [44]. Fentanyl, lidocaine, and ketamine association with the previous protocol was compared to dexmedetomidine (loading dose 1 µg/kg followed by 1 µg/kg/h CRI) combined with lidocaine and ketamine in a recent study by Cardozo et al. [47]. Premedication was carried out with acepromazine (0.02 mg/kg IM) and morphine (0.5 mg/kg IM). Meloxicam (0.1 mg/kg SC and 0.1 mg/kg once a day for 3 days) and tramadol (4 mg/kg) were administered after evaluation after 24 h. The study results showed no significant differences in heart rate between groups, while a lower sevoflurane requirement was observed in dogs treated with fentanyl. However, the association with dexmedetomidine led to a lower need for intraoperative blood pressure support. Both groups had good postoperative pain management assessed with GCMPS-SF, with two and zero dogs receiving rescue analgesia, respectively, for the fentanyl and dexmedetomidine group [47].

Beier et al. [48] studied the use of remifentanil in dogs undergoing mastectomy. Dogs were premedicated with morphine (0.5 mg/kg IM) and acepromazine (0.03 mg/kg IM) and assigned to three groups: control, remifentanil low (0.15 µg/kg/min CRI) or remifentanil high dose (0.30 µg/kg/min CRI). Additional postoperative analgesia was administered, including morphine (0.5 mg/kg IM) and meloxicam (0.2 mg/kg IV). Remifentanil had excellent efficiency as part of a balanced anaesthesia protocol with isoflurane requirement reduction in a dose-dependent manner. However, close heart rate and mean arterial blood pressure monitoring are mandatory and intermittent positive pressure ventilation (IPPV) must be available [48]. In another study, the treatment with cisatracurium (0.2 mg/kg IV), a neuromuscular-blocking agent, led to sevoflurane requirement and propofol dosage reduction in dogs undergoing mastectomy. Intraoperative pain was provided by remifentanil hydrochloride (0.5 µg/kg/min CRI) and postoperative analgesia was provided by buprenorphine (20 µg/kg IV) [49].

### 3.2. Transdermal Analgesia

Transdermal administration of analgesics offers a compelling option for long-term pain relief, minimising the issues associated with chronic parenteral or oral use [50]. Cicirelli et al. [51] compared the fentanyl patch effects to postoperative tramadol (5 mg/kg SC) in dogs undergoing ovariectomy alone or with mastectomy. The fentanyl patches were placed for 72 h on the necks of dogs 24 h before surgery (<10 kg = 25 µg/h; 10–20 kg = 50 µg/h; 20–30 kg = 75 µg/h; 30–40 kg = 100 µg/h). Premedication was performed with dexmedetomidine (3 µg/kg IM) and methadone (0.25 µg/kg IM) and all dogs received postoperative meloxicam (0.20 mg/kg SC). The evaluation of postoperative pain was conducted using NAS and the GCMPS-SF. The study showed that both protocols are effective in postoperative pain management. However, the authors advise to not use fentanyl patches as pre-emptive analgesia during surgery because of the individual variables influencing its absorption [51]. Similarly, another study evaluated the analgesic efficacies of buprenorphine administered with transdermal patches or intravenously in dogs undergoing unilateral mastectomy. Dogs were divided into two groups: one received buprenorphine by patch application in the upper third of the thorax 40 h before surgery (5–6 µg/kg/h); the other one received preoperative buprenorphine intravenously (20 µg/kg), repeated every 6 h for the next 24 h. All dogs received carprofen (4 mg/kg SC) 1 h after surgery. Buprenorphine was found to be insufficient to control the intraoperative nociception stimulus, but may be more helpful in controlling postoperative pain, considering that all dogs in the study did not need rescue analgesia (pain assessed with UMPS and GCMPS-SF) [52]. Therefore, the patch application could be useful in pain management in biting dogs or those who refuse food post-surgery, with the same efficacy as the injectable route, reducing stress related to drug administration. These results are in agreement with other authors who have confirmed that the route of administration influences the analgesic efficacy of buprenorphine in dogs and its bioavailability [53,54].

Furthermore, patches allow the maintenance of a constant drug plasma concentration. Repeated boluses do not allow the maintenance of a stable plasma concentration, which could lead to an incomplete analgesic plateau. Nevertheless, adequate plasma plateau achievement is slower with the transdermal application [51,52].

Considering the above, a multimodal analgesic approach seems to be more effective in dogs undergoing mastectomy. NSAIDs have an established use in the management of postoperative pain in veterinary medicine. The protocols associating drugs such as meloxicam or robenacoxib lead to less need for rescue analgesia. The association of tramadol and meloxicam could be a possibility [26], combined with drugs acting on different levels of the nervous system (e.g., local anaesthetic). The synergic action on different levels of the nervous system is also exploited in protocols using ketamine, lidocaine and fentanyl [44,47]. Fentanyl has great potential in intraoperative and postoperative pain management [43,44,47,51]. Transdermal analgesic administration via patches seems to be promising [51,52]. However, the patch must be applied several hours before surgery to ensure adequate drug blood plateau. This requires more interaction with the animal and greater caution from the owner to prevent the patch from being swallowed. Additionally, the result, especially intraoperative, is not guaranteed, as it is influenced by the degree of absorption.

**Table 1 animals-15-01214-t001:** Papers evaluating systemic analgesic protocols in dogs undergoing mastectomy.

Type	Paper	Drugs and Study Design	Pain Scale
Systemic Administration	Nakagawa et al. [21]	Meloxicam + Butorphanol Vs. Butorphanol	Serum cortisol levels
	Crociolli et al. [22]	Gabapentin + Meloxicam Vs. Placebo + Meloxicam	GCMPS ^1^ + VAS ^2^
	Friton et al. [25]	Robenacoxib Vs. Placebo	GCMPS-SF ^3^
	Texeira et al. [26]	Tramadol Vs. Tramadol + Meloxicam Vs. Tramadol + Dipyrone	GCMPS ^1^ + VAS ^2^
	Uscategui et al. [31]	Methadone Vs. Tramadol	UMPS ^4^
	Uscategui et al. [33]	Methadone	UMPS ^4^
	Sarrau et al. [35]	Morphine + low-dose Ketamine Vs. Morphine + high-dose Ketamine Vs. Morphine	VAS ^2^ + 4A VET pain scale ^5^
	Soares et al. [38]	Ketamine + Lidocaine + Maropitant Vs. Ketamine + Lidocaine	DIVAS ^6^ + NCS ^7^ + GCMPS-SF ^3^
	deMoura et al. [43]	Ketamine + Fentanyl Vs. Ketamine Vs. Fentanyl	GCMPS-SF ^3^ + MNT ^8^
	Marques et al. [44]	Fentanyl Vs. Fentanyl + Lidocaine + Ketamine	GCMPS-SF ^3^
	Cardozo et al. [47]	Fentanyl + Lidocaine + Ketamine Vs. Dexmedetomidine + Lidocaine + Ketamine	GCMPS-SF ^3^
	Beier et al. [48]	Low-dose Remifentanil Vs. High-dose Remifentanil Vs. Control	-
	Interlandi et al. [49]	Cisatracurium Vs. Control	-
Transdermal Analgesia	Cicirelli et al. [51]	Fentanyl patch Vs. Tramadol SC	NAS ^9^ + GCMPS-SF ^3^
	Galosi et al. [52]	Buprenorphine patch Vs. Buprenorphine IV	UMPS ^4^ + GCMPS-SF ^3^

^1^ GCMPS = Glasgow Composite Measure Pain Scale; ^2^ VAS = Visual Analogue Scale; ^3^ GCMPS-SF = Short-Form Glasgow Composite Measure Pain Scale; ^4^ UMPS= University of Melbourne Pain Scale; ^5^ 4A VET pain scale = French Veterinary Association for Anaesthesiology and Analgesia pain scoring scale; ^6^ DIVAS = Visual Interactive and Dynamic Analogue Scale; ^7^ NCS = Numerical Classification Scale; ^8^ MNT = Mechanical Nociceptive Threshold; ^9^ NAS = Numeric Analogue Scale.

## 4. Local and Loco-Regional Analgesia

Papers included in this review evaluating local and loco-regional analgesic protocols and techniques in dogs undergoing mastectomy are summarised in Table 2.

### 4.1. Local Infiltration

The incisional infiltration of a local anaesthetic is defined as the direct injection of a drug, such as lidocaine, bupivacaine, levobupivacaine, or ropivacaine, into the surgical field to support multimodal analgesia. This technique is relatively simple, safe, and cheap and has proven effective in managing intra- and postoperative pain, reducing the use of additional analgesics [55,56]. Surgical site infiltration is usually performed through a syringe connected to a needle. However, there are needle-free devices such as the Comfort-in^®^ (GamasTECH, Catania, Italy). This works as a spring-loaded infusion system and creates a fluidic needle releasing the drug at high speed, penetrating the skin [57]. Comfort-in^®^ was used in dogs undergoing mastectomy surgery with good results. All subjects included in the study were premedicated with dexmedetomidine (2 µg/kg IM) and tramadol (4 mg/kg IM). Before surgery, the incision line was infiltrated with lidocaine (4 mg/kg) with an insulin syringe (capacity 1 mL—27 G) or with a Comfort-in^®^ device, compared to a control group that did not receive local infiltration. The postoperative analgesia scores, assessed throughout the composite pain scale (CPS) and Colorado Pain Scale (CSU-CAPS), and oxidative inflammatory stress levels were lower in dogs receiving local infiltration with the Comfort-in^®^ device without requiring intra- or postoperative rescue analgesia [58]. Wound Soaker Catheters (WSCs) could also be used for subcutaneous infiltration. Although different complications are described in the literature, including surgical site infection [59], WSC use was safe in mastectomy surgery in dogs without any complications or bacterial growth reported [60]. Specifically, dogs were premedicated with acepromazine (30 µg/kg IM) and methadone (0.3 mg/kg IM repeated every 4 h IV) and treated with meloxicam (0.2 mg/kg IV) after surgery. WSC was placed into the subcutaneous space between the subcutaneous tissue and the fascia along the entire length of the surgical wound and removed aseptically after 3 days. Owners administered bupivacaine throughout WSC at 1–2 mg/kg/6 h. Additionally, dogs received carprofen (2 mg/kg/12 h for 5 days PO) and tramadol (3 mg/kg/12 h for 3 days PO). Only two of eighteen patients received rescue methadone (0.1 mg/kg IV) after pain evaluation with GCMPS-SF [60].

Vilhegas et al. [61] studied the analgesic effects of botulinum toxin type A (BoNT-A) subcutaneous injection. Dogs were divided into a group receiving BoNT-A (7 IU/kg diluted to a final volume of 10 mL) and a control group receiving an equivalent volume of saline solution. All dogs received fentanyl (3 mg/kg IM) before both treatments that were performed 24 h before surgery. One millilitre was administered subcutaneously at one point of each mammary gland using a 25 G needle. Before surgery, dogs were sedated with acepromazine (0.03 mg/kg IM) and morphine (0.3 mg/kg IM) and treated with meloxicam (0.2 mg/kg IV plus additional dose 0.1 mg/kg PO after extubation for up to 10 days). Dogs in the BoNT-A group showed lower pain scores with VAS and modified-GCMPS with fewer subjects requiring rescue morphine (0.5 mg/kg IM). Therefore, the authors concluded that BoNT-A administration could be helpful in multimodal approaches in dogs undergoing mastectomy [61]. This seems to be related to the BoNT-A role in blocking the release of neuropeptides at the nociceptive nerve endings reducing peripheral and central sensitisation [62,63]. Additionally, local BoNT-A application resulted in an analgesic effect that remained even after mammary gland removal, which is probably associated with a possible axonal transport distant from the injection site [61,64].

Infiltration of wide body areas could also be provided by tumescent anaesthesia (TA) inoculating large volumes of a solution containing local anaesthetic combined with vasoconstrictors [65]. Due to the diluted nature of the solution, large amounts of the anaesthetic solution may be safely used with a low risk of local anaesthetic-induced toxicity [66], reducing intraoperative bleeding [67] and leading to hydro-dissection of tissues which facilitates surgery [68]. Tumescent anaesthesia is typically performed using the Klein cannula, which has a large diameter and several lateral holes, guaranteeing dispersion of the solution. Additionally, the blunt tip protects from vascular lesions or thoracic and abdominal penetration [69]. Credie et al. [70] applied TA in dogs submitted to mastectomy. All dogs were sedated with acepromazine (0.05 mg/kg IM) combined with meperidine (2 mg/kg IM). Animals received either TA or fentanyl (2.5 µg/kg IV). The tumescent solution was prepared with 250 mL of lactated Ringer’s solution, 40 mL of 2% lidocaine (concentration of 2.75 mg/mL with a total dose of 41.25 mg/kg) and 0.29 mL of adrenaline (1 mg/mL). A total of 15 mL/kg of solution was administered at a refrigerated temperature of 4 °C. The TA was performed with the Klein cannula inserted immediately cranial and caudal to the thoracic and inguinal mammary glands, respectively. After surgery, all dogs received meloxicam (0.1 mg/kg SC). Eight of the ten dogs included in the fentanyl group received a fentanyl bolus. Dogs treated with TA showed less intraoperative bleeding and shorter mammary gland excision times. Additionally, TA improved postoperative pain, assessed with dynamic and interactive visual analogue scale (DIVAS), GCMPS, University of Melbourne Pain Scale (UMPS), VAS and von Frey filament test, without any adverse signs or lidocaine plasma concentration compatible with toxicity [70]. Even if lidocaine is the most frequently used drug [67], other local anaesthetic agents can be used. Ropivacaine has lower toxicity and a greater duration of effect [71]. It was used in tumescent anaesthesia in dogs undergoing mastectomy comparing two different concentrations: 0.1% ropivacaine (216.4 mL of lactated Ringer’s solution + 33.6 mL of 0.75% ropivacaine + 0.5 mL of epinephrine) and 0.05% ropivacaine (233.3 mL of lactated Ringer’s solution + 16.7 mL of 0.75% ropivacaine + 0.5 mL of epinephrine). Premedication was performed with acepromazine (0.04 mg/kg IM) and morphine (0.4 mg/kg IM). The tumescent solution was administered at a 15 mL/kg volume at a temperature ranging from 8 to 12 °C with the Klein cannula. Postoperative pain was assessed by UMPS, modified CMPS and von Frey filaments. The study demonstrated that TA with ropivacaine provided satisfactory postoperative analgesia with no difference in duration between concentrations, maintaining the ropivacaine plasma concentration below the toxic level [72]. Del Lama Rocha et al. [73] compared the effect of TA with 0.1% lidocaine or 0.1% ropivacaine. The tumescent solution with lidocaine was prepared by adding 12.5 mL of 2% lidocaine and 0.5 mL of adrenaline to 237.5 mL of lactated Ringer’s solution. The ropivacaine solution was prepared with 25 mL of 1% ropivacaine and 0.5 mL of adrenaline diluted in 225 mL of lactated Ringer’s solution. The tumescent solution was administered at a 15 mL/kg volume after cooling to 8–12 °C. Dogs were premedicated with chlorpromazine (0.3 mg/kg IM) and meperidine (3 mg/kg IM). The two solutions provided equivalent postoperative analgesia for at least 12 h with no significant differences in pain scores assessed with GCMPS-SF and von Frey filaments. The ropivacaine tumescent solution tended to contribute to a lower sedation degree and a longer duration of postoperative analgesia [73]. The association with lidocaine continuous infusion seems to improve the effects of TA. In a study [74], dogs were premedicated with dexmedetomidine (3 µg/kg IV) and methadone (0.2 mg/kg IV) and assigned to receive TA (12 mL/kg), 2% lidocaine in CRI (loading dose of 2 mg/kg followed by CRI of 100 µg/kg/min) or association of both at the same dosages. The tumescent solution was prepared by adding 40 mL of 2% lidocaine and 20 µg/mL of adrenaline into lactated Ringer’s solution at the temperature of 8 °C. All three protocols showed good results in postoperative pain management, assessed with the Italian version of GCMP-SF (ICMP-SF) [75]. However, the association of continuous infusion of lidocaine and TA led to greater inhibition of the sympathetic stimulating effects with better early postoperative analgesia [74]. Postoperative analgesia is also related to a small gelatinous fraction remaining in the subcutaneous space after mammary gland excision [72,73]. The only adverse effect of TA reported is increased intraoperative hypothermia [70,72,73,74], which could be easily minimised by an electric heating pad and/or radiant heat heating lamp [74,76]. Microwave-heated tumescent solution (37–42 °C) did not prevent the occurrence of intraoperative hypothermia. It did not increase inhaled anaesthetic consumption during surgery. However, greater absorption of the local anaesthetic could not be excluded because the plasma concentration was not evaluated [77]. Generally, although TA has numerous advantages, as previously reported, some limitations should be considered. These include the case of poorly defined tumours, skin infections or ulcerated tumours with the risk of postoperative infection or neoplastic cell seeding [72]. In addition, as reported in other studies in human medicine, it is necessary to use an unground needle to avoid deep lesions, as TA is usually performed without ultrasound control [78].

### 4.2. Epidural Anaesthesia, Intercostal and Paravertebral Block

Epidural analgesia provides effective pain relief for inguinal, perianal, and pelvic limb procedures with minimal side effects that can be implemented in relation to dose, volume, drugs used and speed of injection [79]. However, the extent of its block limits its application in surgeries involving large areas of the thoracic region, such as mastectomy [80]. Thoracolumbar epidural leads to a greater degree of analgesia but the intensity of sympathetic block is not well understood, and adverse effects can occur frequently [81]. Considering the TA application limits, Sanches et al. [82] evaluated the analgesic effect of epidural anaesthesia associated with intercostal block. Dogs were treated with acepromazine (0.03 mg/kg IM) and morphine (0.3 mg/kg IM) and were allocated in two groups. One received TA at the dose of 15 mL/kg (460 mL of Ringer’s lactate solution plus 40 mL of 2% lidocaine and 0.5 mL of epinephrine). The other group was treated with epidural injection in the lumbosacral space with a combination of 2% lidocaine (5 mg/kg) and morphine (0.1 mg/kg), not exceeding a total dose of 7 mg/kg. All dogs received meloxicam (0.2 mg/kg/IV) at the end of surgery. Rescue intraoperative analgesia was most frequently required in dogs that did not receive TA. However, both groups showed adequate analgesia for up to 6 h after surgery using modified GCMPS. Therefore, the authors concluded that the association of epidural anaesthesia and intercostal block could be used as an alternative technique in patients with restrictions for TA [82]. The analgesic area extension of epidural anaesthesia could be reached by combining dexmedetomidine or methadone with levobupivacaine [83]. Specifically, in a study, dogs were premedicated with acepromazine (0.05 mg/kg IM) and midazolam (0.2 mg/kg IM). Dogs were treated with epidural 0.75% levobupivacaine (1.5 mg/kg) alone or in combination with methadone (0.3 mg/kg) or dexmedetomidine (3 µg/kg) in the lumbosacral epidural area (total volume for all treatment of 0.36 mL/kg reached with saline solution). The extent of sensory block was more significant in the groups treated with methadone (7° thoracic to 5° lumbar vertebra) and dexmedetomidine (8° thoracic to 4° lumbar vertebra) than in the group that received only levobupivacaine (12–13° thoracic vertebrae). The motor block duration was longer in dogs treated with dexmedetomidine, which is commonly associated with severe bradycardia. However, all dogs included in the study needed rescue analgesia in the intra- and postoperative period, where the pain was assessed using the Melbourne scale and GCMPS [83]. Herrera-Becerra et al. [84] compared the effect of epidural 0.75% ropivacaine (0.75 mg/kg) associated with morphine (0.1 mg/kg) or xylazine (0.1 mg/kg) or both (final volume 0.35 mL/kg). Dogs were premedicated with acepromazine (0.02 mg/kg IM) and morphine (0.3 mg/kg IM) and treated with meloxicam (0.1 mg/kg IV). All epidural treatments provided adequate intraoperative antinociception and the groups that received morphine had fewer dogs needing rescue analgesia. Pain was assessed with GCMPS-SF for 24 h. The use of xylazine was associated with a longer time to stand. Considering this, the authors advise that epidural ropivacaine and morphine could provide more significant benefit in dogs undergoing mastectomy [84]. The association of epidural ropivacaine and morphine was also investigated by Tayari et al., calculating the drug doses on the occipital–coccygeal length (OCL) (1 mL/10 cm). Dogs were included in the randomised study after premedication with morphine (0.3 mg/kg IM). The study included five groups: a control group, an epidural group receiving 0.5% ropivacaine, and three morphine (0.1 mg/kg) plus ropivacaine groups receiving 0.5%, 0.35%, or 0.25% ropivacaine, respectively. The OCL was measured from the occipital bone to the first coccygeal vertebra. In contrast, the desired extension of the epidural block was measured from the first coccygeal to the first thoracic vertebra. All epidural treatments provided adequate intraoperative analgesia with a reduction in rescue analgesia and inhalant anaesthetic requirements, confirming the dose regimen based on OCL as a safe and practical methodology. Morphine treatment significantly reduced postoperative pain scores measured by GCMPS-SF, while lower ropivacaine concentrations anticipated the resolution of motor block [85].

A thoracic paravertebral block (TPVB) consists of a local anaesthetic injection into the thoracic paravertebral space containing the spinal nerve roots and sympathetic chain [86]. In dogs premedicated with methadone (0.2 mg/kg IV), TPVB with ropivacaine (0.6 mL/kg of a 1% solution diluted with sterile saline solution to a concentration of 0.4%) improved pain control (GCMPS-SF) and recovery in bitches undergoing unilateral mastectomy [87].

### 4.3. TAP Block

Transverse abdominis plane block (TAP block) is a regional anaesthetic technique used to provide local analgesia to the abdominal wall by the ultrasound-guided local anaesthetic injection between the muscle fascia of the abdominal internal oblique and transverse abdominal muscles [88,89]. Two studies analysed the use of TAP block in canine mastectomy surgery [90,91]. In Portela et al. [90], dogs were premedicated with opioid drugs, such as fentanyl or methadone. TAP blocks with 0.25% bupivacaine (0.3–0.35 mL/kg) were performed in two different points: one in the caudal part of the middle abdominal region, cranially to the iliac crest, and a second one in the cranial part of the middle abdominal region, caudally to the last rib (L5–L6 and L2–L3 intervertebral level, respectively). In cases of unilateral mastectomy, the TAP blocks were associated with 9 intercostal blocks with 0.25% bupivacaine (0.03–0.04 mL/kg). After surgery, all dogs received carprofen (2 mg/kg/24 h SC). TAP block provided intraoperative and postoperative analgesia in association with intercostal block for unilateral mastectomy. However, four animals required rescue analgesia perioperatively due to failed intercostal nerve blocks, as nociceptive stimuli were noted during caudal thoracic mammary gland resection. Additionally, Fentanyl was necessary to manage pain following the ligature of structures in the inguinal canal. Indeed, the TAP block only provides regional analgesia of the abdominal wall without involving visceral components [90]. The intercostal block is reached through multiple injections. To avoid this, Teixeira et al. [91] associated Serratus plane (SP) block with TAP block. SP block is a variation of TAP block consisting of drug injection between the serratus anterior and external intercostal muscle [92]. Dogs with mammary tumours were premedicated with methadone (0.5 mg/kg IM). Ultrasound-guided SP and TAP blocks were performed with 0.25% bupivacaine (0.3 mL/kg) injections. After surgery, pain was monitored with the Canine Acute Pain Scale from Colorado State University and all dogs received tramadol (4 mg/kg) and dipyrone (25 mg/kg). The results showed that the combination of both techniques was effective in anaesthetic blocking the thoracic and abdominal wall with good pain management without intra- or postoperative rescue analgesia [91].

Local and locoregional anaesthesia plays a crucial role in multimodal pain management strategies. Specifically, the local infiltration of local analgesic drugs has significant advantages. While the Comfort-in^®^ device has demonstrated effectiveness, it is not suitable for extensive surgical procedures [58]. The number of WSCs used varies depending on the surgery extension. Although no complications related to infections have been reported [60], the use of catheters and their maintenance over several days could pose a risk of infection. In addition, postoperative catheter management and local drug administration is left to the owner, who must therefore be instructed. Among local infiltrations, in our experience, TA offers undeniable surgical and analgesic advantages, with a reduced need for rescue analgesia even when compared to epidural analgesia [70,72,73,74,77,82]. Moreover, the hypothermia risk can be easily controlled and prevented. However, TA application is limited by the clinical presentation of the neoplasm, as already reported. Furthermore, TA can be used in multimodal protocols and be combined with good results with lidocaine in CRI [74]. Epidural anaesthesia has an efficacy limited by the analgesic area extension, which may vary depending on the drugs used and the dosage calculation, volume and speed of injection [79]. Epidural administration of morphine and ropivacaine with a dose regimen based on OCL could be a valuable tool to manage intra- and postoperative pain [85]. TAP block is also a promising technique [90,91], but it alone is not sufficient for pain management during unilateral mastectomy and must be combined with other analgesic techniques. Intercostal blocks, however, involve several injections [90]. The SP block allows thoracic analgesia with a single administration [91]. However, both TAP and SP blocks need a certain expertise and confidence in ultrasonography.

**Table 2 animals-15-01214-t002:** Papers evaluating local analgesic protocols and techniques in dogs undergoing mastectomy.

Type	Paper	Drugs and Study Design	Pain Scale
Local infiltration	Costa et al. [58]	Lidocaine (through Comfort-in^®^ device)	CPS ^1^ + CSU-CAPS ^2^
	Suarez-Redondo et al. [60]	Bupivacaine (through WSC)	GCMPS-SF ^3^
	Vilhegas et al. [61]	BoNT-A	VAS ^4^ + GCMPS ^5^
	Credie et al. [70]	TA (Lidocaine + Adrenaline) Vs. Fentanyl IV	DIVAS ^6^ + GCMPS ^5^ + UMPS ^7^ + VAS ^4^ + von Frey Filament test
	Abimussi et al. [72]	TA (0.1% Ropivacaine + Epinephrine) Vs. TA (0.05% Ropivacaine + Epinephrine)	UMPS ^7^ + CMPS ^8^ + von Frey Filament test
	Del Lama Rocha et al. [73]	TA (Lidocaine + Adrenaline) Vs. TA (Ropivacaine + Adrenaline)	GCMPS-SF ^3^ + von Frey Filament test
	Vullo et al. [74]	TA (Lidocaine + Adrenaline) Vs. Lidocaine CRI Vs. TA (Lidocaine + Adrenaline) + Lidocaine CRI	ICMP-SF ^9^
	Del Lama Rocha et al. [77]	Heated TA	-
Epidural infiltration + Intercostal block	Sanches et al. [82]	TA (Lidocaine + Epinephrine) Vs. Epidural Lidocaine + Morphine + Intercostal block with Lidocaine	GCMPS ^5^
Epidural infiltration	Caramalac et al. [83]	Epidural Levobupivacaine Vs. Epidural Levobupivacaine + Methadone Vs. Epidural Levobupivacaine + Dexmedetomidine	UMPS ^7^ + GCMPS ^5^
	Herrera-Becerra et al. [84]	Epidural Ropivacaine + Morphine Vs. Epidural Ropivacaine + Xylazine Vs. Epidural Ropivacaine + Morphine + Xylazine	GCMPS-SF ^3^
	Tayari et al. [85]	Epidural Ropivacaine Vs. Epidural Ropivacaine + 0.5% Morphine Vs. Epidural Ropivacaine + 0.35% Morphine Vs. Epidural Ropivacaine + 0.25% Morphine Vs. Control	GCMPS-SF ^3^
Thoracic paravertebral block	Santoro et al. [87]	TPVB with Ropivacaine Vs. Control	GCMPS-SF ^3^
TAP block	Portela et al. [90]	TAP with Bupivacaine + Intercostal block with Bupivacaine	GCMPS ^5^
	Teixeira et al. [91]	TAP with Bupivacaine + SP block with Bupivacaine	CSU-CAPS ^2^

^1^ CPS = Composite Pain Scale; ^2^ CSU-CAPS = Canine Acute Pain Scale from Colorado State University; ^3^ GCMPS-SF = Short-Form Glasgow Composite Measure Pain Scale; ^4^ VAS = Visual Analogue Scale; ^5^ GCMPS = Glasgow Composite Measure Pain Scale; ^6^ DIVAS = Visual Interactive and Dynamic Analogue Scale; ^7^ UMPS = University of Melbourne Pain Scale; ^8^ CMPS = Composite Measure Pain Scale; ^9^ ICMP-SF = Italian version of GCMP-SF.

## 5. Unconventional Medicine: Acupuncture

Gakiya et al. [93] evaluated the use of electroacupuncture in pain management in dogs undergoing mastectomy. Specifically, dogs received morphine (0.5 mg/kg IM), electroacupuncture or sham procedure. Electroacupuncture was applied bilaterally to the stomach 36, spleen 6, and gall bladder 34. The numbers that follow the names of acupuncture points correspond to specific points along the body’s meridians, which are distributed on the surface of the body itself. Each point is identified by a code that includes the meridian’s abbreviation and a progressive number that indicates the point’s position along that meridian [94]. The sham procedure was performed similarly to the electroacupuncture, but the needles were inserted into false acupuncture points close to real points. Meloxicam (0.2 mg/kg, IV) was administered to all dogs before the surgical incision. Postoperative pain was measured using a numerical rating scale, while serum cortisol levels were assessed before anaesthetic medication and at 45 min, 1, 3, and 6 h post-extubation. Serum cortisol concentration was not different among the groups, nor was the pain score. However, rescue analgesia was lower in dogs treated with electroacupuncture. The sham procedure was associated with a more evident need for rescue analgesia, underlying the necessity to stimulate specific points to obtain analgesic effects. Thus, the authors concluded that electroacupuncture could be useful to reduce opioid administration in multimodal protocols [93].

Yamamoto New Scalp Acupuncture (YNSA) was studied in dogs undergoing mastectomy [95]. YNSA consists of stimulation of acupuncture points on the scalp associated with other anatomical regions and meridians [96]. Specifically, Bacarin et al. studied YNSA treatment in dogs premedicated with morphine (0.3 mg/kg IM). YNSA was performed through three points: B, D, and E. B point is located 1–1.5 cm lateral to the midline and at the frontal line on the frontalis muscle, and this acupoint is related to different conditions of pain affecting the forelimbs. D point is located on the junction between the upper line of the zygomaticus muscle and its insertion in the scutiform cartilage in the temporal region, thus activating the microregions of the cerebral cortex, triggering an analgesic effect on the lower abdominal region. E point is located above the superior orbit, and it is connected to anatomical structures innervated by afferent fibres originating from T1 to T12 and associated with thoracic analgesia [96]. Morphine (0.1 mg/kg/h IV CRI) and fentanyl were administered during surgery and meloxicam (0.2 mg/kg IV) was administered after intubation. Results showed that intraoperatively supplemental analgesic need was lower in dogs receiving YNSA. Postoperative pain evaluation using interactive VAS (IVAS) recorded from 0.5 to 1 h post-extubation lower pain scores in dogs treated with YNSA. However, postoperative pain assessed with GCMPS-SF and the need for rescue analgesia showed no significant differences between YNSA and the control group. Therefore, the authors concluded that YNSA is useful as a complementary therapy to manage intraoperative pain [95].

Papers evaluating acupuncture protocols are reported in Table 3.

Even if complementary and alternative medicine is not defined as scientifically evidence-based medicine [97], acupuncture is becoming popular in veterinary medicine [98]. However, its use requires a deep knowledge of Oriental Medicine. Further studies are needed to validate its effectiveness in pain management during mastectomy.

## 6. Limitations of the Review

This review has limitations that should be noted.

The first limitation concerns the correlation of pain related to different surgical procedures. Mastectomy procedures can vary from regional mastectomy to unilateral mastectomy to bilateral total mastectomy. This leads to different pain levels [7,8]. Most of the papers included in the review are based on unilateral mastectomy surgeries [21,26,31,33,38,43,44,47,48,49,51,52,61,70,73,74,77,82,83,84,85,87,90,91,95]. However, some studies [22,25,35,58,60,93] have included regional mastectomies. This emphasises that effective approaches addressed in these papers may not be as effective when applied to more radical surgery. Only one study [72] included unilateral or bilateral mastectomies.

Another limitation concerns the difficulty of comparing papers using different pain scales. The pain assessment scales used in the various studies are summarised in Table 1, Table 2 and Table 3. Although many studies use the GCMPS and its short form—both moderately validated—other studies depend on scales less commonly employed in veterinary medicine. Specifically, the 4AVET pain scale and UMPS are only partially validated and only for specific surgical procedures, such as orthopaedic surgery and ovariohysterectomy, respectively [99]. Accurately comparing studies can be challenging. For instance, one study found that tramadol alone was effective in managing postoperative pain [26], while another study reported it was not as effective [31], even though the surgical procedures were similar. The discrepancies in their findings could be related to the use of different pain scales, which complicated the interpretation of the results.

Another limitation lies in the variability of the postoperative observation period across studies.

These considerations highlight the importance of standardising and validating pain assessment scales in relation to the type of surgery, as well as the duration of the observation period.

## 7. Conclusions

Managing perioperative pain in female dogs undergoing mastectomy is a significant challenge in veterinary oncology. Uncontrolled pain affects the animal’s well-being and can also enhance tumour spread due to immunological changes triggered by perioperative stress. Although the traditional use of opioids is common, concerns about their potential adverse effects are growing. Therefore, multimodal analgesic protocols are increasingly recommended. These protocols combine medications with different mechanisms of action to maximise pain relief while minimising side effects. Systemic analgesia is essential for managing pain during and after surgery and includes medications such as NSAIDs, opioids, α2-agonists, and systemically administered local anaesthetics. In addition, local analgesia has emerged as an effective strategy for managing perioperative pain. Techniques like peripheral nerve blocks and the infiltration of local anaesthetics, including transversus abdominis plane (TAP) blocks and epidural analgesia, offer more targeted pain relief, reducing the reliance on opioids and their associated side effects. Additionally, TAP blocks, patches and acupuncture show promise for pain management, but their full potential remains to be explored. The choice of protocol depends not only on the patient but also on the experience of the operator, who may or may not choose to use a multimodal protocol, bearing in mind that some drugs may be used as the sole strategy for intra- and post-operative pain management. 

## Figures and Tables

**Figure 1 animals-15-01214-f001:**
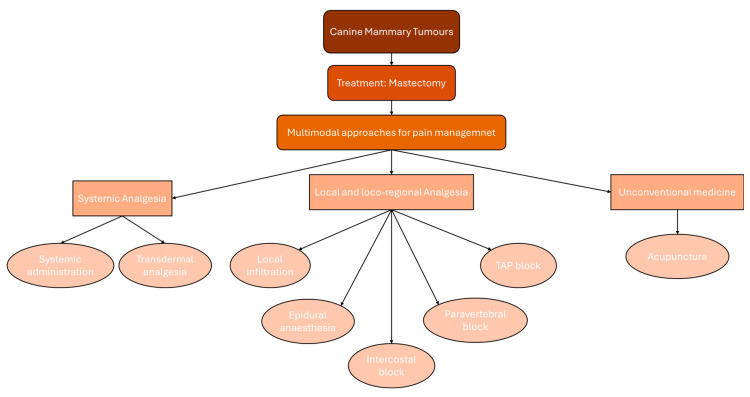
Schematic representation of analgesic administration techniques in dogs undergoing mastectomy.

**Table 3 animals-15-01214-t003:** Papers evaluating acupuncture protocols and techniques in dogs undergoing mastectomy.

Paper	Drugs and Study Design	Pain Scale
Gakiya et al. [93]	Morphine IM Vs. Electroacupuncture Vs. Sham procedure	Numbering rating scale + serum cortisol level
Bacarin et al. [95]	YNSA Vs. Control group	IVAS ^1^ + GCMPS-SF ^2^

^1^ IVAS = Interactive VAS; ^2^ GCMPS-SF = Short-Form Glasgow Composite Measure Pain Scale.

## Data Availability

Dataset available on request from the authors.

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
