# Peer review of "Perioperative Pain Management for Mastectomy in Dogs: A Narrative Review"

_animals, 2025, doi:10.3390/ani15091214_

Round 1
Reviewer 1 Report
Comments and Suggestions for Authors
For this narrative review authors searched reports on analgesic methods used and assessed in oncologic mastectomy procedures in dogs. The searches resulted in a representative sample of publications in the field; however, the review is somewhat incomplete because it lacks or contains only little interpretative summaries, critical analyses of the reported data and/or discussions on potential limitations of the reviewed studies. Instead, the reader receives useful information but limited to (numerical) data of each study; although the tables, summarizing drugs used, are illustrative, what is missing or is insufficient in this reviewer’s view, are summarizing statements, critical consensus, enriched by the authors’ interpretation. This could or should be found as summarizing paragraphs at the bottom of each section. That is, the authors could recommend (or discourage the use of some) analgesic protocols based on the up-to-date and most relevant published evidence, integrated by their own opinion and experience.
M&M – exclusion criteria: omitted were “articles where pain assessment correlated to the surgical technique” this is confusing and warrants explanation. Perhaps there is some misunderstanding, but the key point is that (post-)surgical pain and analgetic treatment are clearly correlated with the extent and invasiveness of surgery. So, why excluding those papers ? In the contrary, there should be a paragraph or a discussion on the limitation of evaluating and comparing analgesic protocols for “mastectomies”, in general; there are partial mastectomies of single or multiple mammary complexes, unilateral or bilateral, there are total mammary chain excisions (warranted or not is not the topic here) – all eliciting different levels of pain and discomfort, and likely requiring different pain relieving protocols – all this is not (easily) extractable from papers, but should be discussed as (inevitable) shortcoming when analysing published data. A further confounding factor is the problem of comparing pain assessed by the different pain scores applied in different studies: perhaps unavoidable, but to be considered when analysing treatment protocols. Last but not least, I strongly suggest evaluating whether the screened study protocols follow the principles of evidenced based medicine; that is, in specifics, to be critical with results and recommendations of alternate medicine-based analgesic protocols. For instance, it might be potentially biased when citing reports of analgetic effects of acupuncture in mastectomy, if such studies were not based on sound scientific methods; therefore, and especially in this context, an interpretation and a critical analysis of M&M and data acquisition is warranted to avoid undue and possibly “painful” effects to our patients.
Please find attached a Word file containing in-line comments and annotations (Review function in Word). I am recommending publication pending revision, considering the comments above and the suggestions in the enclosed file.

Author Response
Dear reviewer,
please find attached the reply to your comments.

Reviewer 2 Report
Comments and Suggestions for Authors
Dear Authors,
Thank you for the great manuscript. It is very well organized. This manuscript will be of use to many clinicians.
However, I have a few questions. Also, there are some parts that may need to be revised.
Abstract
Line19: ‘In conclusion, a multimodal approach, combining different analgesic strategies, is essential to ensure the well-being of dogs undergoing mastectomy, minimising side effects and improving postoperative recovery. ‘
How can you say this? For example, we have this sentence in 175 ‘The association of ketamine (bolus 0.5 mg/kg and intra and postoperative CRI 20 μg/kg/min) and fentanyl (bolus 5 μg/kg and intraoperative CRI 5 μg/kg/hour and postoperative CRI 2 μg/kg/hour) showed no significant differences when compared to groups treated with one of the two drugs’. This paper states that the effect of one drug or two drugs is the same. Do we need to go multimodal to reduce side effects in all cases?
Line28: ‘Tramadol offers good analgesia alone’
This conflicts with what is written in Line 136 ‘However, tramadol alone was inadequate for pain control in a study by Uscategui et al’ . Therefore, may need to rephrase. And please read the paper below.
Donati PA, Tarragona L, Franco JVA, Kreil V, Fravega R, Diaz A, Verdier N, Otero PE. Efficacy of tramadol for postoperative pain management in dogs: systematic review and meta-analysis. Vet Anaesth Analg. 2021 May;48(3):283-296. doi: 10.1016/j.vaa.2021.01.003. Epub 2021 Feb 10. PMID: 33745825.
Line31: ‘maropitant enhances pain management ‘
Kinobe RT, Miyake Y. Evaluating the anti-inflammatory and analgesic properties of maropitant: A systematic review and meta-analysis. Vet J. 2020 May-Jun;259-260:105471. doi: 10.1016/j.tvjl.2020.105471. Epub 2020 May 17. PMID: 32553233.
Hill R. NK1 (substance P) receptor antagonists--why are they not analgesic in humans? Trends Pharmacol Sci. 2000 Jul;21(7):244-6. doi: 10.1016/s0165-6147(00)01502-9. PMID: 10871891.
Based on the above two papers, we may need to rephrase.
Introduction
Line44: ’Mammary gland tumours are the most common hormone-dependent neoplasia in female dogs. ‘
Need a reference.
Line47: ‘Although mastectomy in dogs is associated with a relatively low morbidity, it is considered an invasive surgery that may cause moderate to severe pain.’
Need a reference.
- Systemic Analgesia
Line 238: ‘Buprenorphine showed the same efficacy regardless of the administration route.’
Need a reference. And, the paper below disagrees with your statement.
Steagall PV, Ruel HLM, Yasuda T, Monteiro BP, Watanabe R, Evangelista MC, Beaudry F. Pharmacokinetics and analgesic effects of intravenous, intramuscular or subcutaneous buprenorphine in dogs undergoing ovariohysterectomy: a randomized, prospective, masked, clinical trial. BMC Vet Res. 2020 May 24;16(1):154. doi: 10.1186/s12917-020-02364-w. PMID: 32448336; PMCID: PMC7245774.
Enomoto H, Love L, Madsen M, Wallace A, Messenger KM. Pharmacokinetics of intravenous, oral transmucosal, and intranasal buprenorphine in healthy male dogs. J Vet Pharmacol Ther. 2022 Jul;45(4):358-365. doi: 10.1111/jvp.13056. Epub 2022 Apr 21. PMID: 35445748; PMCID: PMC9543267.
Table 1, Table2, Table3
These tables are very organized, but usually such tables contain the results from each article. Having the results next to the Pain Scale would be easier for the reader to understand and would help in choosing a protocol at the time of surgery. For example, ‘Effective’ ‘Not effective’, etc.
- Local and Loco-Regional Analgesia
Line 262: ‘This technique is relatively simple, safe, and cheap and has proven effective in managing intra- and postoperative pain, reducing the use of additional analgesics. ‘
You state that this technique is safe, but I always hesitate to use this technique for tumors. I am concerned about the seeding you have mentioned in other parts of your article(Line 363). For skin tumors I am always afraid of that risk because I can't exactly see that hidden tentacle and my technique could touch that tentacle. If you have a paper that says the possibility of seeding is small and safe, please let me know. I also think that information would relieve many readers.
Line 368: Epidural analgesia provides effective pain relief for inguinal, perianal, and pelvic limb procedures with minimal cardiorespiratory effects.
Need a reference. And, as far as I know, this statement is not accurate. I believe that the coverage of analgesia depends on the type and volume of drugs.
Conclusions
Line 506: ‘However, relying solely on systemic analgesics may not provide adequate pain control. ‘ & Line 513:’ A personalised approach that integrates systemic and locoregional analgesia could establish the gold standard for improving the well- being of veterinary cancer patients, minimising the negative impacts of surgery and perioperative pain.
Why can you say this? Do you basically do multimodal on dogs undergoing mastectomy? For example, a few hospitals I know of avoid the use of fentanyl. They just use ketamine, then, lidocaine as needed.
Thank you for the excellent manuscript. It is very well written and will be helpful to the dogs and clinicians who need this surgery. I believe the manuscript would be even better if you could revise it with my input. Thank you.
Author Response
Dear Reviewer,
please find attached the reply to your comments.

Round 2
Reviewer 1 Report
Comments and Suggestions for Authors
Thank you for the revision which seems complete and adequately responds to all comments and suggestions; there are a few oversights (type errors, sentences eg.line 151 that need to be corrected, the word "eventually" has different meaning in English (just delete), unexplained abbreviations and few others; I have included the pdf file where you can find these areas which need attention; they are highlighted in red (within the yellow parts of the text). Corrected these few items, this reviewer recommends publication.

Author Response
Please find attached the reply to your suggestions.

Reviewer 2 Report
Comments and Suggestions for Authors
Thank you for the revisions you made to the manuscript in response to the comments I previously gave to you. This paper will contribute to many veterinarians and patients.
Author Response
Dear Reviewer,
thank you for your contribution to this paper. We really appreciated it.